# Liquid Biomarkers for Pediatric Brain Tumors: Biological Features, Advantages and Perspectives

**DOI:** 10.3390/jpm10040254

**Published:** 2020-11-27

**Authors:** Sibylle Madlener, Johannes Gojo

**Affiliations:** Comprehensive Center for Pediatrics, Department of Pediatrics and Adolescent Medicine, Medical University of Vienna, 1090 Vienna, Austria; johannes.gojo@meduniwien.ac.at

**Keywords:** pediatric cancer, brain tumor, liquid biopsy, cerebrospinal fluid, ddPCR, biosensor, next-generation sequencing (NGS), miRNA

## Abstract

Tumors of the central nervous system are the most frequent solid tumor type and the major cause for cancer-related mortality in children and adolescents. These tumors are biologically highly heterogeneous and comprise various different entities. Molecular diagnostics are already well-established for pediatric brain tumors and have facilitated a more accurate patient stratification. The availability of targeted, biomarker-driven therapies has increased the necessity of longitudinal monitoring of molecular alterations within tumors for precision medicine-guided therapy. Nevertheless, diagnosis is still primarily based on analyses of the primary tumor and follow-up is usually performed by imaging techniques which lack important information on tumor biology possibly changing the course of the disease. To overcome this shortage of longitudinal information, liquid biopsy has emerged as a promising diagnostic tool representing a less-invasive source of biomarkers for tumor monitoring and therapeutic decision making. Novel ultrasensitive methods for detection of allele variants, genetic alterations with low abundance, have been developed and are promising tools for establishing and integrating liquid biopsy techniques into clinical routine. Pediatric brain tumors harbor multiple molecular alterations with the potential to be used as liquid biomarkers. Consequently, studies have already investigated different types of biomarker in diverse entities of pediatric brain tumors. However, there are still certain pitfalls until liquid biomarkers can be unleashed and implemented into routine clinical care. Within this review, we summarize current knowledge on liquid biopsy markers and technologies in pediatric brain tumors, their advantages and drawbacks, as well as future potential biomarkers and perspectives with respect to clinical implementation in patient care.

## 1. Introduction

### 1.1. General Information to Pediatric Brain Tumors, Diagnosis, Prognosis, Treatment and Clinical Benefit of Liquid Biopsy

Tumors of the central nervous system (CNS) are the most frequent solid tumor type in children and adolescents and exhibit the highest mortality rate of pediatric cancer types [1]. Although the overall prognosis of pediatric brain tumors is more favorable than in adults, their high contribution to cancer-related mortality in children can be attributed to the high aggressiveness of certain subtypes including high-grade glioma (HGG), medulloblastoma (MB), ependymoma (EPN) and other rare tumor types such as atypical teratoid rhaboid tumors (ATRT) or embryonal tumors with multilayered rosettes (ETMR) [2]. These entities account for approximately one third of pediatric brain tumors and are currently treated with intense therapeutic regimens which, however, are still not curative in a substantial proportion of cases [1,2]. These therapies are mostly based on radiotherapy as well as chemotherapy and differ in between entities [2]. Importantly, these highly toxic therapies may also result in substantial therapy-associated sequelae including amongst others risk of secondary malignancies, endocrinopathies and impaired neurological development [3]. Current classification is based on the “2016 World Health Organization Classification of Tumors of the Central Nervous System” [4] which implemented molecular markers in the diagnosis of pediatric brain tumors. Thus, tissue biopsy and magnetic resonance imaging (MRI) are the current gold standard methods for pediatric brain tumor diagnosis [5] as well as for longitudinal monitoring during therapy and follow-up. However, tissue biopsies are mostly taken at a single time point, mostly at primary diagnosis of the tumor, and only represent the molecular composition of the primary tumor. Furthermore, follow-up by MRI is feasible to monitor tumor size and recurrence but lacks the detailed information of molecular alterations during tumor evolution. In addition, the availability of targeted therapy has increased the need for longitudinal monitoring of molecular changes within tumors in order to precisely guide anti-cancer treatment. Hence, there is a compelling need for the indirect assessment of tumors at certain time points via less-invasive tumor markers which can be gained from body fluids of the patient [6,7]. The most proximal source for liquid biopsy for pediatric brain tumors is the cerebrospinal fluid (CSF) followed by peripheral blood including its components serum/plasma and urine [8] (Figure 1). Diverse biological specimens are detectable within these body fluids by multiomics approaches and may be specific for different brain tumor types.

### 1.2. Liquid Biopsy in General

Over the last decade substantial progress has been made in the understanding of the molecular pathogenesis of pediatric brain tumors through multi-omics screening methods [9,10,11,12,13,14,15,16,17] and the discovery of novel and sensitive screening methods reveals a great opportunity for liquid biomarkers. Multiple different molecular structures have been identified in the liquids of pediatric brain tumor patients and raised the interest of the scientific field. These biomarkers include circulating tumor cells (CTCs) [18,19], circulating tumor DNA (ctDNA) [7,18,19], cell-free DNA (cfDNA) [6,18,19,20,21], circulating proteins [6,8,19,20], extracellular vesicles and exosomes [5,18,22,23,24,25,26], micro-RNAs (miRNAs) [5,19,27,28,29,30,31,32], long non-coding RNA (lncRNA) [33] and other genetic alterations [28,34,35,36,37,38]. Detection of these biomolecules may offer advantages compared to surgical interventions. Nevertheless, there are still some challenging limitations for the clinical application with respect to standardized isolation protocols and sensitive methods. These pitfalls will be addressed in the second chapter.

#### 1.2.1. Circulating Tumor Cells

Circulating tumor cells were described and documented in the blood of a patient with metastatic disease for the first time in 1869 in Australia [39,40]. Moreover, detection of tumor cells in CSF via cytology is routinely used for staging of certain pediatric brain tumor types. Any detection of circulating tumor cells in body fluids from patients provides a direct evidence of tumor spreading and metastasis [19] and may be used as an early biomarker to predict tumor diagnosis, progression and its dissemination [41]. Through shedding into the CSF or blood vessels, the collection and isolation of these tumor cells from the liquids is a simple and less-invasive intervention for tumor monitoring and clinical treatment decisions. Another important advantage of circulating tumor cells over ctDNA and cfDNA is the longer half-life of these cells which ranges from 1–2.4 h [42] in taken blood or CSF samples. Therefore, detection of circulating tumor cells might be a good and almost stable real-time marker for tumor surveillance [41].

#### 1.2.2. Cell Free/Circulating Tumor DNA (cf/ctDNA)

DNA fragments originating from cells outside of the blood circulation are known to be found in the bloodstream and other bio-fluids from patient and healthy humans. They are released from the cells into the fluids by several cellular processes, such as apoptosis or necrosis, respectively [40,43]. Usually, cf/ctDNA appears in combination with other biomolecules in the form of a protein complex. The size of ctDNA ranges from 70 to 200 bp, a longer DNA fragment than normal cfDNA [43]. They are also released randomly into microvesicles or exosomes and carried to other cells [44]. These short DNA sequences offer the perfect transcript for the direct detection of any genetic alterations, mutations or single nucleotide variants by using highly sensitive methods which are now available and will be described in detail within the next part of the article. With respect to adult brain tumors, Li et al. demonstrated that CSF derived ctDNA better reflect the mutations in driver genes compared to plasma ctDNA in glioblastoma patients. This result is partly explained by the high concentration of ctDNA in CSF and its low concentration in plasma because of the blood brain barrier [45]. However, the clearance of these small DNA sequences in the bloodstream and CSF are not complete understood and need to be investigated in more detail. Nonetheless, there are many open questions before ct/cfDNA will be used as a routine biomarker including stability, degradation and concentration in the liquids as well as specific isolation kits for these small DNA fragments.

#### 1.2.3. Extracellular Vesicles/Exosomes

Several studies have demonstrated that multiple cells, including tumor cells and tumor stem cells, release extracellular vesicles and exosomes and internalize exosomes secreted from other cells [46,47,48,49,50]. They originate from multivesicular bodies and are discharged via p53- regulated exocytosis [51]. During their composition they pack active and selective several cellular structures, such as miRNA, mRNA, lncRNA, proteins, DNA, or lipids [44,52], respectively, and deliver them to other cells [53]. Exosomes represent a cargo tool between cellular complexes and have the ability to cross the blood brain barrier [54] as shown in different xenograft models [55,56,57]. This approach offers a broad spectrum for diagnostic and prognostic biomarker studies. Furthermore, the exosomal miRNA content is a hallmark of tumor cell types and the incorporation of miRNAs, respectively, is highly specific [5,58] to cancer cells. During past years, treatment abilities of exosomes loaded with specific active biomolecules like miRNAs have been tested in different clinical trials for cancer immunotherapy [59].

#### 1.2.4. Micro-RNA (miRNA)

Micro-RNAs constitute an evolutionary highly conserved class of non-coding RNAs that play key roles in the regulation of gene expression. They are genome-encoded short, single-stranded approximately 22 nucleotide-long RNA molecules that silence or contribute to degradation of mRNA by binding to 3′-untranslated regions of mRNAs. Their complete function is not yet fully understood, but recent research has shown small, non-coding microRNAs to be master regulators of cellular processes, and expression patterns have revealed their potential as diagnostic, prognostic, and treatment response biomarkers [60]. Furthermore, miRNAs appear to be extremely stable outside the cells and can be easily detected in extracellular space [61], extracellular vesicles and exosomes [5], or in other types of body fluid such as blood, urine, breast milk, CSF, cyst fluids [62,63,64]. Therefore, miRNAs are particularly suitable as clinical biomarkers for early detection of disease, monitoring cancer progression and therapeutic response [63].

## 2. Sample Handling and the Major Challenge and Pitfall for Sufficient Liquid Biopsy Detection

As already mentioned, the extensive interest in detailed knowledge from the molecular biology of pediatric brain tumors and the trend to precision medicine has fueled development of novel molecular methods in pathology. Technological developments and advances of extremely sensitive and reliable methods have improved the feasibility of analyzing liquid biopsy samples from brain tumor patients. However, there are still certain challenges and pitfalls to avoid false positive or negative results. In this chapter, we will discuss the major pitfalls of liquid biopsy.

The quality and handling of the liquid biopsy sample is an important point in the pre-analytical phase. The stability of liquid biopsy markers and structures are mainly affected by several steps during the pre-clinical workflow, such as what kind of blood collection tube is used followed by the storage temperature and time-to-processing as well as centrifugation speed [65]. All these factors could cause a reduction of stability and concentration of these easy to obtain, but fragile biomarkers. Therefore, a standard operating procedure is urgently needed for the pre-analytical phase to reach reliable and good comparable results within different routine labs. 

### 2.1. Circulating Tumor Cells 

Beside the pre-analytical pitfalls in handling of body fluids, circulating tumor cells are represented in an extreme low concentration in liquid biopsy samples in early stage tumors [66]. Especially in pediatric brain tumor patients, drawing large volumes of liquids is almost impossible. Therefore, an adaption of the circulating tumor cell isolation is inevitable. In 2016, Gorges and colleagues tested an in vivo device for the isolation of circulating tumor cells from a lung cancer patient. They validated the CellCollector from GILUPI (GMBH, Potsdam, Germany) that enables the isolation of circulating tumor cells direct from the arm vein of the patient. The wire from the CellCollector surface is modified and coated with anti-EpCam antibodies and only EpCam positive cells will bind. During the incubation of 30 min the modified wire is exposed to approximately 1 L of blood, thereby an increasing concentration of circulating tumor cells was achieved [67]. This new device might be the promising tool to isolate sufficient circulating tumor cells in all kind of tumor patients. 

### 2.2. Cell Free/Circulating Tumor DNA (cf/ctDNA)

The major challenge of using cf/ctDNA for liquid biopsies is their extremely low abundance in serum. Depending on the tumor stage, the ctDNA ranges between 0.01% to more than 10% in total isolated cfDNA [68]. To increase the stability and avoid the degradation of these rare DNA structures, specific coated collection tubes are needed. In 2019, Sese and colleagues tested and validated cell-free DNA collection tubes for their usage in liquid biopsy. They obtained blood from a lung cancer patient and shipped the processed samples to the Vall d’Hebron Hospital by two different methods in parallel: (1) cell free DNA blood collection tubes from Roche, shipped at room temperature; and (2) in standard K_2_EDTA blood collection tubes with centrifuged plasma, shipped frozen on dry ice. Thereafter, they analyzed the Epidermal growth factor receptor (EGFR) mutation status and demonstrated the same high significance and concordance of the cell free DNA tubes shipped on room temperature as the current method (K_2_EDTA blood collection tube). In summary, the simple handling of the cell-free DNA collection tubes for the shipping and storage on room temperature for up to 5 days indicates a preferable method for future clinical settings in routine labs [69]. The isolation method of cell-free nucleic acid is another crucial pre-analytical step for liquid biopsy. In 2020, Kerarchian et al. demonstrated a significant difference in the methods according to sensitivity in the downstream analysis. They compared the silica membrane isolation gold standard for cf/ctDNA isolation from liquids with the “selective capture of ctDNA on magnetic beads” method (SCC-MAG) [70]. Beside the increased detection values in polymerase chain reaction (PCR)-based methods in the downstream workflow, this new technique was capable of isolating sufficient ctDNA from hemolytic samples. The SCC-MAG technology can be applied to any specific DNA sequence. In their respective study, the authors analyzed the specific mutations of *KRAS* in ctDNA isolated from patient serum samples.

### 2.3. Exosomes

In general, exosomes are extreme stable in the bloodstream and would be the perfect liquid biopsy tool. Their content includes miRNAs, proteins and, recently, researchers discovered the presence of dsDNA from the parental tumor cell [71]. Certainly, there are still problems in the clean filtration and isolation of the exosomes. Beside their tendency to bind on plasma proteins, they are extreme sticky to common laboratory plastic thereby reducing the isolated concentration [72]. Even the time from the blood draw to the isolation procedure negatively manipulates the number of collected exosomes. Hence, more progress in the development of sensitive and reproducible methods is needed before exosomes could be introduced into clinical routine.

### 2.4. Micro-RNA (miRNA)

Micro-RNAs have been found to be stable within extracellular vesicles and exosomes isolated from CSF and plasma samples. Balzano and colleagues addressed the question of the stability of miRNAs in frozen vs. fresh plasma samples [73]. There, they evaluated the miRNA concentration of fresh plasma samples compared to frozen samples after 6 and 12 months of storage at −80 °C. In addition, they included and evaluated plasma samples stored at −80 °C and collected in 1999, 2002, 2003, 2009 and 2010. They observed no significant differences of the miRNA values between the fresh and 6-month frozen samples. Interestingly, only one miRNA (miR126-3p) showed a lower concentration in the years 2010, 2009, 2003, and 2002 compared to the fresh plasma samples. All other miRNAs had no significant differences in these years. They explained that the stability of miRNAs is related to the absence of the AU sequence in the seed and tail miRNA regions [73] and is not generally connected to time.

## 3. Sensitive Methods

In this chapter we discuss the latest findings of sensitive techniques for the detection of markers in liquid biopsies 

### 3.1. Next-Generation Sequencing (NGS)

Next-generation sequencing (NGS)-based technologies provide high-throughput screening of the biopsy samples and improved accuracy with respect to mutation allele frequency (MAF), gene fusions or DNA amplification [74,75]. As previously described, NGS methods can be applied to targeted panel systems for detection of targeted ctDNA mutations and indels. These panels are namely Safe-Sequencing System (Safe-SeqS), Cancer Personalized Profiling by deep sequencing (CAPP-Seq), Ion Torrent and Tagged-Amplicon deep sequencing (TAm-Seq) [74]. Gale et al. published in 2018 an improved version of the TAm-Seq based NGS and named it eTAm-Seq technology. They investigated the InVision liquid biopsy platform with eTAm Seq and increased the detection of low-frequency mutations in cfDNA. The novel enhanced system can detect MAF as low as 0.25% with a sensitivity of 94% compared to the normal TAm-Seq (MAF ~2% and 97% sensitivity) [75,76]. Targeted panel sequencing represents low-cost and high sensitivity to point mutation in liquids, but lacks detection of whole genome sequencing (WGS) or whole exome sequencing (WES). These methods provide a complete profile of the tumor DNA or exome DNA including point mutations, indels, rearrangements, copy number variations (CNV) and DNA amplification. However, WGS and WES are pricy and both sequencing technologies are less sensitive and need a high amount of sample input. This essential aspect impedes their usage for liquid biopsy applications in early tumor diagnosis when ctDNA concentration is known to be low in the patient liquids.

### 3.2. Digital Polymerase Chain Reaction (dPCR) and Digital Droplet PCR (ddPCR)

Over the past decade, substantial progress has been made in the development of more sensitive PCR-based methods. The new explored dPCR and ddPCR [35,75] methods are ultrasensitive and low-cost techniques with respect to sequencing analysis. Furthermore, they are able to detect allele variants or targets with low abundance in samples that are below the limit of conventional quantitative PCR (qPCR) methods. In the last few years, several studies demonstrated this detection limit of rare events (mutation, CNV, miRNAs or other cell-free analytes [6,18,35,37,38,77]) in liquid biopsy samples. However, PCR-based methods are limited to screening already known targets and by sample input and speed. 

### 3.3. Electrochemical Biosensors

During the last few years, the electrochemical biosensor technology has made substantial progress in the sensitive detection of nucleic acids in liquid biopsy samples [78,79]. Until 2015, the detection of cf/ctDNA in liquids with electrochemical biosensors was still challenging. Das et al. improved the technology by combining a specific electrochemical clamp assay [80] with new surface nanostructure, and DNA probe design. They mixed isolated ctDNA from patient samples with peptide nucleotide acids (PNA) clamps for main known *KRAS* mutation and wild type and hybridized the sample. The clamp sequence for the mutation (target) of interest was excluded in the mixture. Thereafter, the heterogeneous solution including ctDNA and binding partners were applied to the PNA probe-modified microsensor. The target sequence lacking clamp partner hybridized to the PNA probes on the modified nanostructured microchip. Beside the specific PNA design, Das and colleagues coated the Au nanostructure chip with a thin layer of Pd to obtain nanostructured microelectrodes and reached a higher sensitivity of the binding and measured finely the differential pulse voltammetry (+target or −target). They tested this new assay design successfully in undiluted serum samples of lung and melanoma cancer patients [80].

Recently, another breakthrough in the development of electrochemical microfluidic biosensors was achieved. Bruch et al. in cooperation with our group developed a microfluidic chip combined with the novel Clustered Regularly Interspaced Short Palindromic Repeats (CRISPR/Cas13a) technology [81,82]. This system features a highly sensitive and selective detection tool for numbers of different targets. In a first study it allowed the detection and quantification of tumor miRNA-19b in serum samples from medulloblastoma patients without any amplification step. In addition, the complete process of analyzation was performed in less than 4 h and for successful measurements, a minimum sample volume of 0.6 µL was used. All results were validated with the current qPCR-based gold standard for miRNA detection. The next step of Bruch and colleagues will be the development of a multiplexed CRISPR/Cas13a biosensor for analyzing miRNAs in patient samples at once. This improvement will provide a multiple accessible and low-cost tool for fast detection of nucleic acids in liquid biopsy samples.

## 4. Potential Liquid Biopsy Markers for Specific Brain Tumor Entities

Having high potential for improved patient management, several studies have already investigated different methods for liquid biomarker detection in pediatric brain tumors. The following section and Table 1 summarize the current status for each entity and provide an outlook for potential molecular aberrations which could be leveraged as biomarkers in the future. 

### 4.1. Glioma

Low- and high-grade gliomas cover a wide clinical spectrum ranging from long-term treatment and follow-up to highly aggressive tumors with limited treatment options and short overall survival. Molecular analyses have revealed distinct genomic drivers and uncovered inter- as well as intratumoral heterogeneity within this entity. Importantly, the majority of low-grade gliomas is driven by alteration of the BRAF-pathway (*BRAF*-fusions, mutations) which are promising candidates for liquid biopsy tumor detection. These alterations (e.g., *BRAFV600E*-muation) may also be found in a subset of high-grade glioma. Indeed, first reports demonstrate that detection of *BRAFV600E* in liquid biopsies is feasible [35,38,89]. However, sensitivity in blood ranges from 25–50% suggesting that further improvement of the techniques is necessary [38]. This is of importance as targeted therapies against *BRAFV600E* exist and have been shown to effectively inhibit tumor growth in these patients [90,91]. Thus a sensitive detection method could aid in therapy decision and monitoring of the respective cases. Another major oncogenic driver found in gliomas are mutations within histones (*H3F3A, HIST3B*) which are frequently found in the midline and are thus not suitable for resection or re-biopsies. Consequently, the possibility of other means for monitoring tumor development appears important. *H3K27M*-mutations have been shown to be reliable detected in both blood and CSF [34,35,37,83]. Whereas the sensitivity of *H3K27M*-detection by ddPCR was about 90% in both CSF and plasma [35], panel sequencing appears to be more sensitive in CSF [83]. In general, ctDNA was found to be higher in CSF as compared to blood but, nonetheless, Panditharatna et al. could show that plasma-derived ctDNA levels are related to therapy and recurrence [35]. These first studies on brain stem glioma have also demonstrated that other mutations in genes such as *ACVR1, IDH1* or *TP53* can be reliably detected [35,83]. Apart from point mutations, gene fusions (e.g., *NTRK1/2/3, ALK, ROS1*) have been found in a subset of pediatric glioma and could also serve as biomarkers for the respective targeted therapeutics which are already approved. However, studies investigating this aspect in pediatric brain tumors are still lacking. Last, copy number aberrations (CNA) such as amplification of *MYC/PDGFRA* or deletion of *CDKN2A/PTEN* are frequently found in high-grade glioma and could also serve for disease monitoring. Regarding miRNAs, one study has already shown that miR-21, miR-15b, miR-23a, and miR-146b are elevated in the serum of pilocytic astrocytoma patients and that the level correlated with tumor size and therapy response [84]. Sensitivity and specificity of this approach were 86% and 100%, respectively. With respect to protein detection a study investigating elevated levels of bFGF and TIMP3 in urine demonstrated 98% specificity for pilocytic astrocytoma whereas levels in the urine of medulloblastoma, high-grade glioma or non-tumor patients were negative.

### 4.2. Medulloblastoma

Molecular subgroups of medulloblastoma are well established and can be robustly determined via their distinct DNA methylation patterns [92]. Metastatic medulloblastoma usually spreads within the ventricular system making CSF a very promising biofluid for disease monitoring. Consequently, analysis of DNA methylomes in patient body fluids could serve as a potential biomarker. Moreover, certain medulloblastoma subtypes harbor somatic alterations such as *TERT* Promoter-, *CNNB1, PTCH1* or other mutations which could be leveraged for liquid biopsy monitoring [15]. In addition, characteristic CNA including monosomy 6, *GLI2*-, *MYCN*-, of *MYC* amplifications are also good candidates [93]. Apart from genomic aberration, we have recently shown that CSF of medulloblastoma patients is also characterized by a distinct lipid profile which may be used as a biomarker [85]. Corroboratively, prostaglandin D2 synthase (PDG2S) has been shown to be reduced in CSF of medulloblastoma patients [86]. In contrast, insulin-like growth factor binding-protein 3 (IGFBP3) has been shown to be enriched [87]. With respect to miRNAs, we and others have already shown that medulloblastomas exhibit specific miRNA signatures [94,95] but their feasibility for liquid biopsy detection warrants further investigation.

### 4.3. Ependymoma

Similar to medulloblastoma, ependymoma subgroups identified by DNA methylation patterns are well-established biomarkers [96] and thus represent an ideal candidate for liquid biopsy biomarkers. Apart from genome-wide methylation patterns, *TERT* promoter methylation has also been suggested as a potential biomarker in ependymoma which could be detected via ctDNA [97]. With respect to supratentorial ependymoma, gene fusions of *RELA*- and *YAP* are frequent somatic events which harbor potential for detection in body fluids [98]. In some cases even in posterior fossa ependymomas somatic mutations (*H3K27M*) may serve as biomarker [99]. The utility of ddPCR for detection of spinal ependymoma in adult patients has already been investigated but led to mixed results as ctDNA could not be reliably detected [100]. Importantly, selected CNAs such as the gain of chromosome 1q or 5p are also associated with dismal outcomes in ependymoma [97,98] and could also serve as biomarkers.

### 4.4. Rare Entities

Molecular dissection across the landscape of pediatric brain tumors has uncovered numerous novel tumor entities with specific genomic alterations which could be used for liquid biopsy monitoring. Atypical teratoid rhaboid tumors (ATRT) are characterized by mutations or deletion within *SMARCB1* and would be both accessible to liquid biopsy technologies. Moreover, they comprise diverse subtypes which can be determined by DNA methylation analysis [101]. On a protein level, osteopontin has been shown to be elevated in plasma and CSF of ATRT patients and a rise in level is associated with tumor recurrence [87]. 

Embryonal tumors with multilayered rosettes (ETMR) is another rare pediatric brain tumor type which harbors either a characteristic focal amplification of the C19MC-cluster or mutations in DICER1 [102], both genetic events potentially suitable for liquid biopsy analysis via ctDNA.

Next, CNS high-grade neuroepithelial tumor with genetic alterations in either *BCOR* (HGNET-BCOR), *FOXR1* (HGNET-FOXR1) or *MN1* (HGNET-MN1) are newly described subtypes all characterized by distinct molecular alterations [14]. Apart from their specific methylation profile, analysis of genomic aberrations such as internal-tandem duplications in the case of CNS HGNET-BCOR could be suitable for liquid biopsy detection.

## 5. Conclusions

Over the past decade, interest in biological markers isolated from patient body fluids have emerged in the scientific world and have already revolutionized the diagnosis and tumor surveillance of certain entities. However, there are still multiple pitfalls to bypass, beginning with the pre-analytical workflow of the sample processing up to the sensitivity, stability and reliability of the potential biomarkers. The first studies have already shown that this approach is also feasible for pediatric brain tumors providing potential biomarkers and further development is currently ongoing. However, the costs of the equipment for liquid biopsy techniques are still expensive and the establishment of a routine liquid biopsy lab in each center currently appears uneconomical. Consequently, centralized liquid biopsy analysis would be a feasible strategy to ensure broad availability of these methods. In addition, certain techniques such as electrochemical biosensors also harbor the potential for decentralized diagnostics in the future. Taken together, liquid biopsy could offer a more precise and comprehensive diagnosis and monitoring of pediatric brain tumors in the future, in particular in guiding personalized anti-cancer therapies, but further efforts are required until liquid biopsies can be integrated into the clinical routine and patient care.

## Figures and Tables

**Figure 1 jpm-10-00254-f001:**
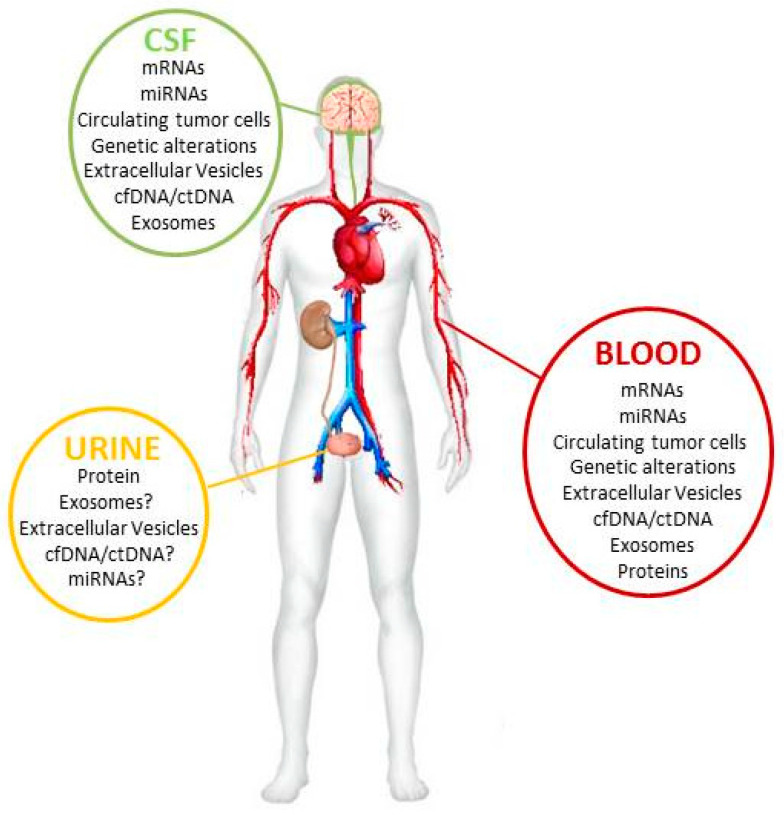
Schematic overview.

**Table 1 jpm-10-00254-t001:** Published and potential liquid biopsy biomarkers.

	Biomarker	Source	Method	Literature
Ependymoma				
Genomic	DNA methylation, gene fusions (RELA-, YAP), CNA (1q, 5p, CDKN2A)	blood/CSF	*NGS, ddPCR, methylation array*	
Glioma				
Genomic	*DNA methylation,* Mutations (e.g., BRAF, H3F3A, HIST3B), *CNAs* (e.g., *CDKN2A*), *Fusions* (e.g., *NTRK*)	blood/CSF	*NGS, ddPCR, methylation array*	[34,35,37,83]
miRNA	miR-21, miR-15b, miR-23a, miR-146b	blood/*CSF*		[30,84]
Proteomics	bFGF, TIMP3	Urine	ELISA	[8]
Medulloblastoma				
Genomic	DNA methylation, somatic mutations (CNNB1, TERT promoter, PTCH1 and others), CNA (monosomy 6, Gli2, MYCN, MYC)	*blood/CSF*	*NGS, ddPCR, methylation array*	
lipidomics	lipid profiles in CSF	CSF	lipidomics	[85]
protein	PDG2S, IGFBP3	CSF	ELISA	[86,87]
miRNA	*miRNA profiles*	*blood/CSF*	*NGS*	
Rare tumors				
ATRT				
Genomic	DNA methylation, Mutations (SMARCB1), DNA methylation		*NGS, ddPCR, methylation array*	
Protein	Osteopontin	blood/CSF	ELISA	[88]
ETMR	DNA methylation, CNA (C19MC), Mutations (DICER1), miRNA		*NGS, ddPCR, methylation array*	
HGNET-BCOR	BCOR internal tandem duplication		*NGS*	

CSF, cerebrospinal fluid; NGS, next-generation sequencing; ddPCR, digital-droplet polymerase chain reaction (PCR); ELISA, enzyme-linked immunosorbent assay.

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
