# Peer review of "Liquid Biomarkers for Pediatric Brain Tumors: Biological Features, Advantages and Perspectives"

_jpm, 2020, doi:10.3390/jpm10040254_

Round 1

Reviewer 1 Report

The Authors present an interesting paper titled: " Liquid biomarkers for pediatric brain tumors: biological features, advantages and perspectives" very clear written and with  a clear content easy to understand and very understandable.

I don't have concern in the presented sections of the manuscript: clear Introduction, clear description of the tests, clear adequate application to the brain tumors.

Only a minor request: 1. give a short note about the costs of the development of liquid biopsy techniques into clinical routine 2. where do they are applied ? everywhere, in any Center? or is it necessary to centralized the samples?

I consider for applicants useful these information.

Author Response

Response to Reviewer 1 Comments

We thank the reviewer for general positive evaluation of our manuscript and for the helpful comments.

Point 1: give a short note about the costs of the development of liquid biopsy techniques into clinical routine. Where do they are applied? everywhere, in any Center? or is it necessary to centralized the samples?

Thank you for your comment on our submitted manuscript. We addressed your question and comments in the conclusion of our manuscript as followed:

"However, the costs for the equipment for liquid biopsy techniques are still expensive and the establishment of a routine liquid biopsy lab in each center would be uneconomical. The future will be a centralized lab for each country or region to analyze the samples."

We hope we could answer all your points and we are looking forward to hearing from you.

Reviewer 2 Report

Please find a commented version of the manuscript attached. In general, this review is not well structured and could be improved following the comments in the manuscript and a better structure such as: 

1. Introduction
1.1 general information to pediatric brain tumors (Glioma, Medulloblastoma, Ependymoma and rare entities), diagnosis, prognosis, treatment and the clinical benefit of LB.

1.2 liquid biopsy in general 

1.2.1 CTCs,  1.2.2 ctDNA, 1.2.3 EVs, 1.2.4 miRNA 

2. Sample handling and pitfalls (discriminate between the molecular entities)
2.1 CTC
2.2 ctDNA
2.3 EVs
2.4 miRNA

3. Sensitive methods
3.1 NGS 
3.2 ddPCR 
3.3 electrochemical biosensors 

4. Liquid biopsy in specific pediatric brain tumors
4.1 Glioma
4.2 Medulloblastoma
4.3 Ependymoma
4.4 Rare entities

5. Conclusions

Author Response

Response to Reviewer 2 Comments

We thank the referee for the thorough and highly constructive review of our manuscript. We have now restructured the manuscript according to the reviewers’ suggestions and have further implemented the suggestions and comments raised in the commented version of the manuscript.

Point 1: In general, this review is not well structured and could be improved following the comments in the manuscript and a better structure such as: 

1. Introduction
1.1 general information to pediatric brain tumors (Glioma, Medulloblastoma, Ependymoma and rare entities), diagnosis, prognosis, treatment and the clinical benefit of LB.

1.2 liquid biopsy in general 

1.2.1 CTCs,  1.2.2 ctDNA, 1.2.3 EVs, 1.2.4 miRNA 

2. Sample handling and pitfalls (discriminate between the molecular entities)
2.1 CTC
2.2 ctDNA
2.3 EVs
2.4 miRNA

3. Sensitive methods
3.1 NGS 
3.2 ddPCR 
3.3 electrochemical biosensors 

4. Liquid biopsy in specific pediatric brain tumors
4.1 Glioma
4.2 Medulloblastoma
4.3 Ependymoma
4.4 Rare entities

5. Conclusions

Response 1: We appreciate your constructive comments on our submitted manuscript and adapted it according to your structural suggestions. We subdivided the introduction point 1 into 1.1 for general information to pediatric brain tumors, prognosis, treatment, and the clinical benefit of liquid biopsy and 1.2 liquid biopsies in general followed by the four subgroups 1.2.1 CTC, 1.2.2 cf/ctDNA, 1.2.3 Exosomes, and 1.2.4 miRNA.

For the second part of the manuscript, we separated the entities and described their challenges and pitfalls for successful liquid biopsies. Now, the table of content in the second chapter is as followed: 

2. Sample handling and the major challenge and pitfall for sufficient liquid biopsy detection

2.1 CTC, 2.2 cf/ctDNA, 2.3 Exosomes and 2.4 miRNA

Further changes according to commented PDF-Version:

  • The abstract was revised now more clearly stating the relevance of liquid biopsy for pediatric brain tumors
  • A paragraph summarizing the clinical and molecular landscape of pediatric brain tumors was added in the introduction setting (page 1 and 2 of revised manuscript)
  • The sections on circulating tumor cells and exosomes were thoroughly revised following the suggestions of the reviewer (pages 4 and 5 of the revised manuscript)
  • Table 1 was reformatted following the reviewers suggestions
  • language editing (whole manuscript)

We hope that our revised manuscript is now suitable for publication and we are looking forward to hearing from you

Round 2

Reviewer 2 Report

Clear structure, reads well, no further concerns. 

I recommend acceptance.